# RoboTube: Learning Household Manipulation from Human Videos with Simulated Twin Environments

**Haoyu Xiong**[*1,2]  **Haoyuan Fu**[*3]  **Jieyi Zhang**[3]  **Chen Bao**[3]  **Qiang Zhang**[4]

**Yongxi Huang**[3]  **Wenqiang Xu**[1,3]  **Animesh Garg**[5]  **Cewu Lu**[†1,3]

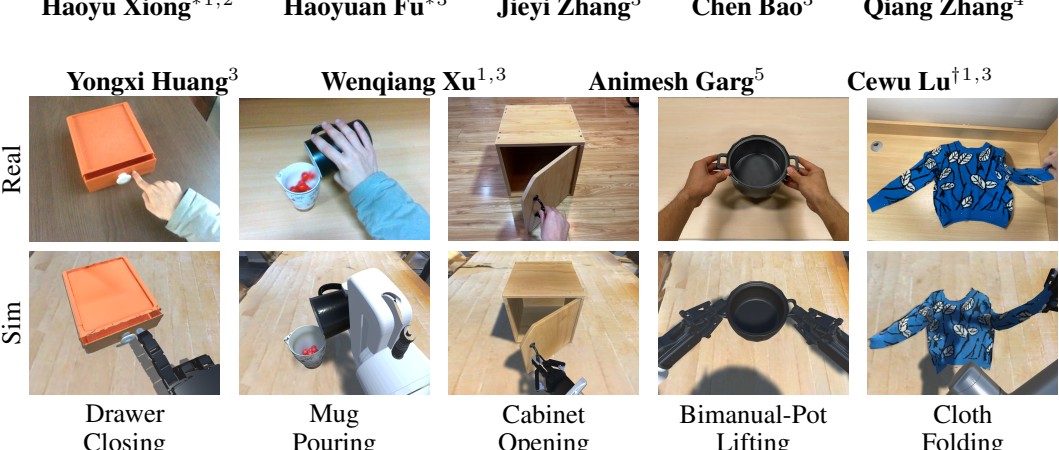

Figure 1: **RoboTube** covers a wide range of household manipulation tasks. RoboTube constructs a human video dataset and a suite of simulated twin environments for reproducible research. The first row shows the examples of the real-world video frames; the second row shows the simulated twin environments.

**Abstract:** We aim to build a useful, reproducible, democratized benchmark for learning household robotic manipulation from human videos. To realize this goal, a diverse, high-quality human video dataset curated specifically for robots is desired. To evaluate the learning progress, a simulated twin environment that resembles the appearance and the dynamics of the physical world would help roboticists and AI researchers validate their algorithms convincingly and efficiently before testing on a real robot. Hence, we present RoboTube, a human video dataset, and its digital twins for learning various robotic manipulation tasks. RoboTube video dataset contains 5,000 video demonstrations recorded with multi-view RGB-D cameras of human-performing everyday household tasks including manipulation of rigid objects, articulated objects, granular objects, deformable objects, and bimanual manipulation. RT-sim, as the simulated twin environments, consists of 3D scanned, photo-realistic objects, minimizing the visual domain gap between the physical world and the simulated environment. After extensively benchmarking existing methods in the field of robot learning from videos, the empirical results suggest that knowledge and models learned from the RoboTube video dataset can be deployed, benchmarked, and reproduced in RT-sim and be transferred to a *real robot*. We hope RoboTube can lower the barrier to robotics research for beginners while facilitating reproducible research in the community. More experiments and videos can be found in the supplementary materials and on the website: https://sites.google.com/view/robotube.

**Keywords:** Learning from Videos, Video Demonstration Dataset, Real2Sim, Self-supervised Reward Learning, Robotic Simulation Benchmark

## 1  Introduction

Conceptualizing robotic manipulation tasks by diverse *human videos* unlocks the potential to enable general household robots [1–4]. Prior works have made fruitful progress on manipulation tasks such as pick-and-place by learning from offline video datasets [5–9]. As these video datasets facilitate the pioneer exploration of robotic manipulation learning, they have several limitations for further exploration:

---

[*1]Shanghai Qizhi Institute, [2]Carnegie Mellon University, [3]Shanghai JiaoTong University, [4]Tsinghua Univeristy, [5]University of Toronto & Vector Institute, [*] are equal contributors. [†] is the corresponding author.

6th Conference on Robot Learning (CoRL 2022), Auckland, New Zealand.

(1) **Scale up task complexity**. Previous datasets and frameworks in robot learning from videos mainly focus on the simple manipulation tasks, e.g., grasping [5], pushing [6], relocating rigid objects [7], etc. While a practical robotic manipulation system should be able to handle more complex tasks. RoboTube involves household manipulation tasks of articulated objects, deformable objects, granular objects, as well as bimanual coordination.

(2) **Balance Data diversity & relevance**. It is difficult, if not impossible to learn everyday household tasks from video datasets that are collected on a static lab table with limited object instances [7, 10–12]. A robotics-oriented video dataset that contains more diverse content is missing and needed. Recently, research has been conducted on how massive-scale open-world video datatsets from computer vision community [8, 9] contributes to generalization in robotic manipulation tasks [1, 2, 13]. However, as they are not originally designed for robotics, they introduce unnecessary challenges for manipulation tasks with irrelevant or even misleading content. For example, in Ego4D dataset [9], the video frames may have content beyond human manipulation including a crowd in a live concert, human walking, etc. Collecting a video dataset that balances diversity and relevance remains an open problem in this area.

(3) **Function on comparing baselines**. The video dataset alone is not enough. The lack of a standard testing environment paired with the dataset makes the meaningful, reproducible, democratized comparison among different baseline methods extremely hard. For example, [1, 13] both learned reward functions and induced policies from the same something-something dataset [8] but applied the learned models to different robotic experiments. We aim to build a simulated benchmark that would help roboticists and AI researchers validate their algorithms convincingly and efficiently before testing on a real robot, thus, encouraging future research in the area of robot learning from videos.

To address the limitations mentioned above, we introduce **RoboTube** (Fig. 2), a human video dataset of around **5,000** RGB-D video demonstrations and a suite of simulated twin environments.

1) To ensure the task complexity, RoboTube setups 5 task families, namely drawer-closing (*articulated object with prismatic joint*), mug-pouring (*granular object*), cabinet-opening (*articulated object with revolute joint*), bimanual-pot-lifting (*bimanual coordination*), and cloth-folding (*deformable object*).

2) To take the data diversity into consideration, for each task family, we ask 9 demonstrators to conduct the task with diverse but *natural* hand poses upon different objects of the same category which have variations in shapes, materials, and textures. We collect the videos in both clean and cluttered scenes. To support the reproduction and comparisons of different algorithms and enable wider applicability, the RoboTube video dataset contains multiple functionalities. We collect both *successful* (expert video demonstrations) and *failed* (negative video demonstrations) episodes, concerning 50 tasks and 60 objects. Two temporally synchronized video streams are recorded from a first-person viewpoint (FPV) and a third-person viewpoint (TPV).

3) To benchmark the baseline methods, we construct a simulated twin environment, RT-sim, for the tasks and objects. With RT-sim, researchers can make a fair comparison of their approaches with the baseline methods and can validate their algorithms convincingly and efficiently before conducting more complex experiments on real robots.

We conduct extensive experiments to benchmark existing methods on RoboTube. We select three self-supervised reward learning methods, namely goal classifier [14, 15], TCN [4], and XIRL [16], and evaluate them on unseen tasks via reinforcement learning. Besides, we also conduct visual pretraining on RoboTube videos, and use the learned models on RT-sim. The empirical results elucidate that the models learned from the RoboTube video dataset can be transferred, used for policy learning, and benchmarked in RT-sim. Finally, we evaluate the successfully trained models on a real robot, the experiment results show performance consistency on RT-sim and real robot setups.

We summarize our contributions as follows:

1) We identify the issues in existing human video datasets for robot learning, and curate a benchmark, RoboTube, which is designed by jointly considering the human video dataset and the evaluation platform. RoboTube not only introduces more complex tasks with diverse object types, but also supports meaningful, reproducible, democratized comparisons among different baseline methods.

2) We conduct extensive experiments to benchmark existing methods with the RoboTube. The empirical results suggest that the models learned from RoboTube video dataset can be deployed, benchmarked, and reproduced in RT-sim. The real world robotics experiments also show the sim2real transfer ability of RT-sim.

| Dataset Name | Depth Mapping | Multiple Viewpoints | Simulated Twin | Negative Sample | End-Effector | Approx Annotation | Number of Tasks | Dataset Size |
|---|---|---|---|---|---|---|---|---|
| XIRL(real) [16] | ✗ | ✗ | ✗ | ✗ | Diverse | N.A | 1X | 100 videos |
| G-in-W [5] | ✓ | ✗ | ✗ | ✗ | DemoAT | gripper open/close | 1X | 12 hours |
| TCN-pour [4] | ✓ | ✗ | ✗ | ✓ | Human hands | N.A | 1X | 300 videos |
| RLV [3] | ✗ | ✗ | ✗ | ✗ | Human hands | N.A | 2X | 300 videos |
| DexMV [7] | ✓ | ✓ | ✓ | ✗ | Human hands | 6 object models | 3X | 700 videos |
| VIME [6] | ✗ | ✓ | ✗ | ✗ | DemoAT | gripper transition | 2X | 2000 videos |
| **RoboTube** | ✓ | ✓ | ✓ | ✓ | Human hands | **60 object models** | **5X** | **5000 videos** |

Table 1: **Comparison of video demonstration datasets.** We compare the features of RoboTube video dataset with related video demonstration datasets. In this table, DemoAT means demonstration assistive tools. In the number of tasks section, n X meams n groups of tasks.

## 2 Related Work

### 2.1 Offline Datasets for Robotic Manipulation

Leveraging offline datasets to learn diverse manipulation behaviors has been studied by previous researchers.

**Video datasets for perception tasks.** The computer vision community has curated many human-object-interaction (HOI) video datasets for different perception tasks [8, 9, 17]. [1, 13] have proved that it is effective to learn a generalizable reward function from the something-something dataset [8]. R3M [2] also exploited Ego4D [9] to improve efficiency in downstream motor control tasks. Despite the rich prior knowledge that HOI videos have provided, such datasets are not originally designed for robotics. For example, Ego4D dataset [9] contains content beyond human manipulation including crowd in a concert live, human walking, etc.

**Action-included demonstrations for robotic manipulation.** An action-included demonstration usually contains both the visual observation and the corresponding actions of the robots, which provides strong supervision for a robot to learn complex behaviors. Previous works collect demonstrations on a static lab table [10–12]. Recently, several works [18, 19] take an effort to enrich the data diversity and show better generalization ability in imitation learning of everyday household tasks. Despite the tremendous progress in learning from action-included demonstrations has been made, such datasets suffer a key problem: it is time-consuming and expensive to collect everyday household activities by guiding and/or teleoperating a real robot entity. In contrast, one can record videos anywhere and anytime with a portable camera.

**Video-only demonstrations for robotic manipulation.** Consider the issues of other two kinds of datasets, roboticists have also constructed video-only datasets for robotic manipulation [3–7, 16]. These datasets can be divided into two mainstreams: *robot-friendly* video demonstrations [5, 6], *human-friendly* video demonstrations (human videos) [3, 4, 7, 16]. Song et al. [5] propose a robot-friendly interface for collecting video demonstrations anywhere using assistive tools (DemoAT). Besides DemoAT, researchers also propose to collect videos with human hands for robotic manipulation. DexMV [7] conducts a novel pipeline to bridge 3D vision and dexterous manipulation. A more detailed comparison of RoboTube's features to those of related datasets can be found in Table **??**.

### 2.2 Algorithms for Robot Learning from Videos

Endowing robots with the ability to learn skills by simply observing humans has been an emblematic north star problem in robotics [14–16, 20–24]. Several directions have been proposed to achieve this goal:

**Reward learning from videos.** Recent works demonstrate impressive manipulation skills learned from human videos by inverse reinforcement learning [4, 13–16]. For example, previous works [13, 14] train a goal classifier as a reward function on human videos for policy learning. Later, Xie et al. propose DVD [1], a domain-agnostic video discriminator for generalizable reward learning. More recently, XIRL [16] leverages temporal cycle-consistency constraints [25] to learn deep visual embeddings that are aware of task progress.

**Visual pre-training for motor control.** Recent works also discussed how to connect computer vision to policy learning by leveraging self-supervised pre-training. A line of works [2, 26, 27] has shown that pre-trained vision models from diverse real-world data can be effective to improve policy learning. For example, Nair et al. [2] prove that vision-language pre-training on diverse egocentric datasets, e.g., Ego4D [9].

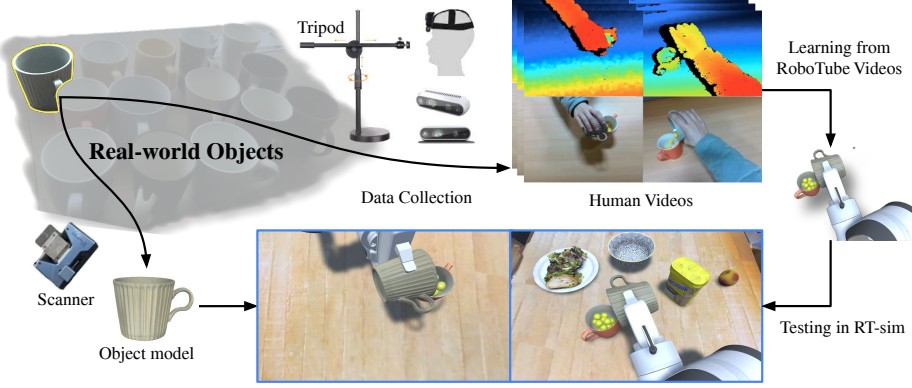

Figure 2: **Overview of RoboTube.** When building the video dataset, we ask demonstrators to collect manipulation video demonstrations recorded by multi-view RGB-D cameras. Meanwhile, we scan the corresponding objects into high-fidelity 3D models and construct simulated twin environments, RT-sim. After learning from the video dataset, we test the learned models in the RT-sim.

# 3 RoboTube Benchmark: Pairing Human Videos with Digital Twins

RoboTube consists of a diverse human video dataset and a suite of simulated twin environments, RT-sim. In this section, we will introduce the ideology of our design choice and the details of the benchmark construction step by step.

## 3.1 Task Definitions in RoboTube.

**Task Family.** With the idea of task complexity in mind, we define 5 task families, with which we hope to go beyond pick-and-place and cover common household manipulation tasks for diverse objects with different levels of complexity. Specifically, the five task families are *articulated object manipulation* (drawer-closing, cabinet-opening), *granular object handling* (mug-pouring), *deformable object manipulation* (cloth-folding), and *bimanual coordination* (pot-lifting).

**Two modes: Stuctured, cluttered.** To ensure data diversity, we set up two task modes for video dataset and RT-sim for each task family. As shown in Fig. 3(a), we design 1) the *structured mode*, where we place only the object on a clean table as the *easy level*, and 2) the *cluttered mode*, the *hard* level, where we place the objects in diverse real-world scenes without intentional clean-up, i.e., muliple distractors exist along with the objects in the scenes.

**Diverse Object Selection** Each task family contains multiple object instances of the same category with variations in colors, shapes, and textures, but consistency in semantics and affordance. There are 10 drawers, 20 mugs, 10 cabinets, 10 pots, and 10 cloths, in total, 60 objects in RoboTube.

**Train-Test Split** We aim to learn a vision-based manipulation policy that generalizes to unseen domains. In RoboTube, we split the objects in each task family into a training set (80%), and a testing set (20%). For the RoboTube video dataset, the testing set is different from the training set videos not only in terms of objects but also scenes, and viewpoints. We encourage the users to train the models with training set videos, and test their performance on testing set videos and RT-sim with unseen objects, viewpoints, and scenes,

## 3.2 Construction of the Video Dataset

We construct a collection of human video demonstrations for robotic manipulation. RoboTube video dataset contains rich functionalities, equipping it with the capability as a benchmark for existing algorithms with different settings. An overview of our RoboTube dataset is shown in Fig. 3.

**Video Collection Setup.** Collecting videos in everyday life with affordable and portable devices boosts the data scale and diversity for robotics. We designed a portable video collection system with two RealSense D435 cameras with a resolution of $640 \times 480$ and a frequency of 30 Hz. During recording, two viewpoints are streamed: one is the first-person perspective from the camera mounted

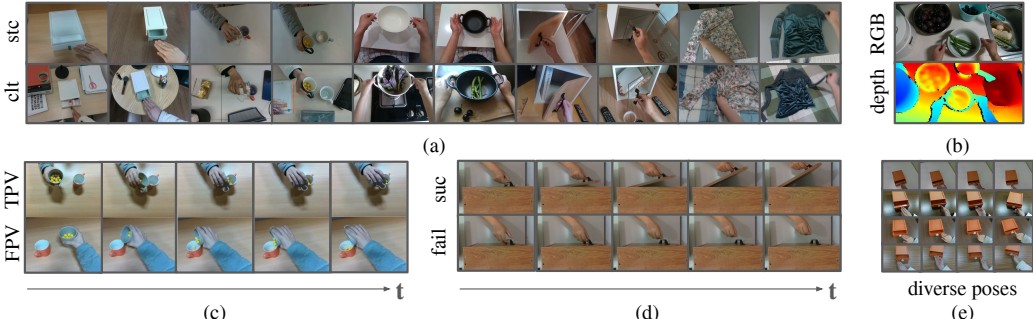

Figure 3: **Preview of RoboTube videos dataset.** (a): RoboTube designs the structured (stc) mode and the cluttered (clt) mode for two levels of task difficulty. The first row shows the structured scenes of drawer-closing, mug-pouring, pot-lifting, cabinet-opening, and cloth-folding tasks. The second row shows the cluttered scenes of the five manipulation tasks. (b): Each frame in the RoboTube video dataset contains an RGB stream and a depth stream. (c): A first-person viewpoint (FPV) camera and a third-person viewpoint (TPV) camera are temporally synchronized. (d): RoboTube video dataset provides both successful episodes and failed episodes for the same task. (e) given the example of the drawer closing task, human demonstrators are required to make diverse poses to complete the tasks.

on the human head, and the other is the third-person perspective from the camera fixed on a tripod placed near the scene. These two streams are temporally synchronized.

**A Video Dataset with Rich Functionalities.** To support the reproduction and comparisons of different algorithms and enable wider applicability, we design multiple functionalities for the RoboTube video dataset. For each task family, we ask 9 demonstrators to conduct the task with diverse but *natural* hand poses upon different objects of the same category which have variations in shapes, materials, and textures. 5,000 video demonstrations with both RGB and depth images are collected in both clean and cluttered scenes. We collect both *successful* (expert video demonstrations) and *failed* (negative video demonstrations) episodes. Two temporally synchronized video streams are recorded from a first-person viewpoint (FPV) and a third-person viewpoint (TPV). We believe it encourages future directions to leverage negative demonstrations and multi-viewpoint videos for self-supervised learning.

### 3.3   RT-sim: Building Realistic Simulation Environments from Real-to-Sim.

To provide an accessible test platform for reproducible research of robot learning from videos, we design RT-sim, a suite of simulation environments *paired* with RoboTube video dataset.

In RT-sim, we take special care to build visually realistic assets. Following the object scanning and annotation procedures in [28], we scan high-fidelity textured mesh models for every single object shown in the video dataset. The visually realistic assets serve as the digital twins of the objects shown in real-world videos. The digital twins can be transferrable to other simulation environments as well. To create realistic everyday household scenes, We import the scenes from Matterport3D [29] and use google object scans [30] as the interactable distractors for cluttered mode. The visualization of the RT-sim cluttered scenes can be found in Fig: 4.

To bridge the gap between the simulation and the real world, visual rendering and physics simulation play important roles. We use uses Unity's underlying physics engine technology which provides photorealistic rendering quality for indoor scenes.

Moreover, RT-sim supports various robots (e.g. Franka, UR5, Kinova-gen3) and grippers (e.g. Allegro Hand, Robotiq 85) for manipulation tasks.To enable robot learning algorithm training, RT-sim provides a standard OpenAI Gym [31] API in Python language. The details of each task family can be found in the appendix.

## 4   Experiments

As mentioned earlier, the mainstream of robot learning from videos adopts self-supervised reward learning. In this section, we benchmark three self-supervised reward learning methods on RoboTube. To validate the real-world generalization, we also deploy the learned reward models on a real robot. More experiments on benchmarking visual pre-training for robotic manipulation on RoboTube can be found in supplementary materials.

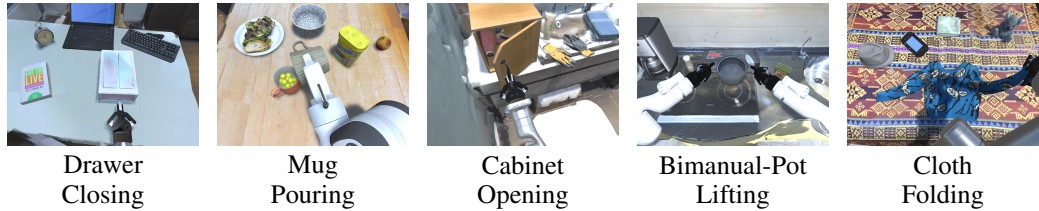

| Drawer Closing | Mug Pouring | Cabinet Opening | Bimanual-Pot Lifting | Cloth Folding |

Figure 4: **Cluttered RT-sim gallery.** We render all cluttered-mode scenes of RT-sim from the first-person viewpoint.

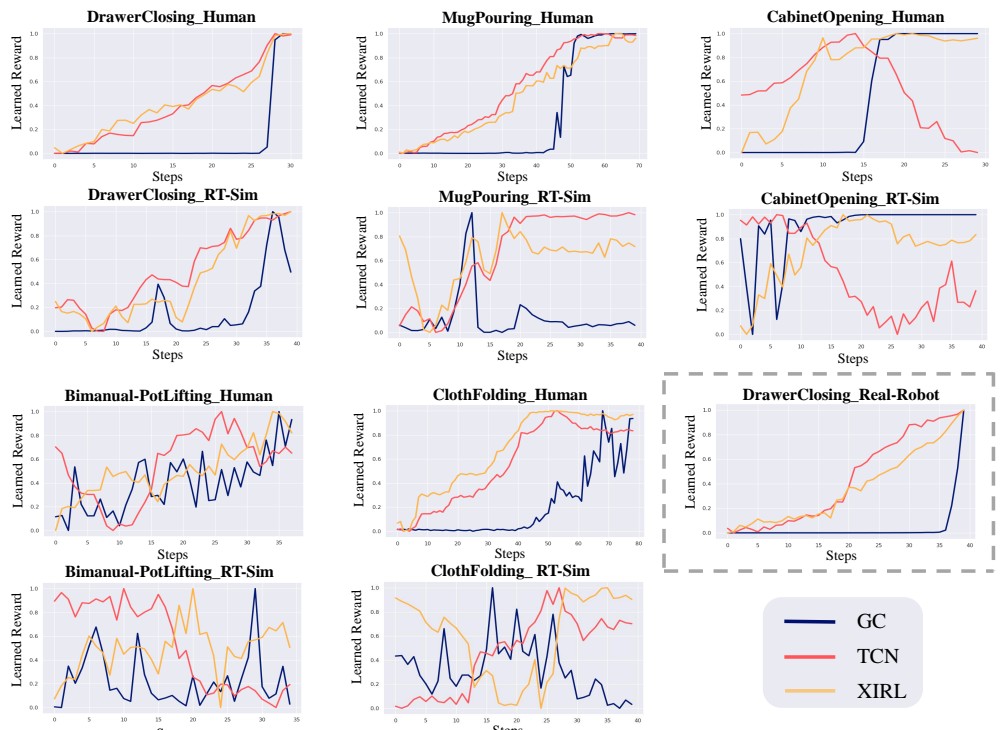

Figure 5: **Performances of the learned reward models.** To test the generalization ability of the learned reward models, we train three self-supervised reward learning models on RoboTube dataset. At testing time, we use two successful video trails for each of all five tasks to predict the reward. For each task, from top to bottom are testing videos of human domain and structured mode in simulated robot domain. We also use a real robot demonstration video, in which Franka successfully finishes drawer closing task, to predict the reward as shown above.

## 4.1 Self-supervised Reward Learning

In this subsection, we benchmark three self-supervised reward learning approaches on RoboTube. We aim to answer the question: can we learn reward models that effectively generalize from the RoboTube video data (real) to RT-sim data (sim)?

Followed by the *1) Goal Classifier (GC)* method used in Automated Visual Instructionfollowing with Demonstrations (AVID) [14] and Learning by Watching (LbW) [15], we create self-supervised signals by considering the last five frames of the video as the positive label and the rest as the negative labels. We use the output probabilities as the reward by simply training a binary classifier. We evaluate the single-view *2)Time Contrastive Network (TCN)* [4] on RoboTube. TCN has been demonstrated effective on several *Imitation from Videos* experiments as a standard baseline, including pouring task [4], multi-stage coffee-making task [4], cross-embodiment video games [16]. We adapt single-embodiment version of *3) Cross-embodiment Inverse Reinforcement Learning (XIRL)* [16] in our experiments. XIRL leverages Temporal Cycle-Consistency Learning (TCC) [25] constraints to learn embeddings that are aware of task progress from offline videos. In TCN and XIRL, we take the negative distance between the current state and goal state in embedding space as a reward function.

|  | DrawerClosing | MugPouring | CabinetOpening | Bimanual-PotLifting | ClothFolding |
|---|---|---|---|---|---|
| *Max Success Rate* |  |  |  |  |  |
| Env-rew | 99.9% ±0.0% | 2.6% ±0.3% | 40.1% ±17.3% | 63.5% ±25.0% | 0.7% ±0.5% |
| GC | 27.1% ±3.5% | 1.7% ±0.5% | 0.3% ±0.3% | 1.3% ±0.3% | 0.0% ±0.0% |
| TCN | 98.2 % ±5.9% | 0.4% ±0.5% | 0.0% ±0.0% | 1.0% ±0.5% | 0.0% ±0.0% |
| XIRL | 100.0% ±0.0% | 3.6% ±1.1% | 11.6% ±2.7% | 0.3% ±0.3% | 0.0% ±0.0% |
| *Goal Metrics* |  |  |  |  |  |
| Env-rew | 0.999 ±0.000 | 1.617 ±0.262 | 33.001 ±10.880 | 0.038 ±0.011 | -0.630 ±0.005 |
| GC | 0.271 ±0.035 | 0.696 ±0.150 | 3.666 ±0.398 | 0.007 ±0.002 | -0.640 ±0.000 |
| TCN | 0.982 ±0.012 | 1.474 ±0.190 | 27.860 ±2.547 | 0.003 ±0.001 | -0.640 ±0.002 |
| XIRL | 1.000 ±0.000 | 1.191 ±0.294 | 58.777 ±5.624 | 0.003 ±0.001 | -0.637 ±0.002 |

Table 2: Max success rate and goal metric. We compare the max success rates and max goal metrics averaged over three seeds of environment reward method and three baseline methods in this table.

Considering the structured mode, we leverage the RGB stream of the RoboTube dataset with third-person viewpoints for reward model training. At testing time, we use two successful video episodes to predict the reward from the learned reward models for each task family. we use the *1) human test video*, a sample from the test set of the RoboTube video dataset, we aim to verify if the reward models can generalize to unseen objects in the human domain. We also investigate whether self-supervised reward learning baselines can generalize to unseen objects in the simulated robot domain by testing on *2) RT-sim test video*, which is a successful video clip generated from RT-sim.

After benchmarking three self-supervised reward learning methods on five task families, as shown in Fig. 5, we find that learned reward models achieve reasonable success on *human test videos* for all five task families. For the testing on *RT-sim test video*, reward predictions produced by XIRL are positively correlated with the testing on *human test videos* for the drawer closing, mug pouring, and cabinet opening tasks. This indicates that XIRL successfully overcome the real-to-sim visual gaps in RoboTube (human videos to RT-sim gap) with the photo-realistic rendering provided by RT-sim. Hence, we prove that models learned from RoboTube video data can be evaluated in the paired simulation environments (RT-sim) smoothly. However, with the increasing task difficulty of bimanual pot-lifting and cloth folding, it still remains challenging for XIRL to accurately predict the rewards on RT-sim test videos. GC has an obvious advantage on articulated object manipulation tasks. However, it fails in other tasks. TCN performs well in drawer closing, mug pouring, and cloth folding, but fails in the other two tasks.

### 4.2 Generalization to Unseen tasks via Reinforcement Learning

We consider the problem of reinforcement learning with pre-trained reward models. Particularly, we are interested in whether the reward models learned from the RoboTube video dataset can be effective for the downstream robotic manipulation tasks in the RT-sim compared with the hand-crafted reward function. We use a low-level state as the input of the Soft-Actor-Critic [32] policy network, and visual observation (an RGB image) as the input of the pre-trained reward model.

We compare the success rates and goal metric averaged over 3 seeds on unseen tasks of the five task families for RL approaches of using hand-crafted reward functions (noted as Env-Rew), and using learned reward models. Note that the objects and background scenes in RT-sim environments are never seen during reward model training and objects are segmented into a training set and a testing set.

We report the success rate and goal metric of the policy learning experiments as shown in Table 2. Based on the observation of the decreasing performance of the RL with Env-rew and learned reward models, the difficulty of the tasks gradually increases from drawer closing to cloth folding. Step-by-step difficulty levels ensure RoboTube includes both easy tasks (drawer closing) and challenging tasks, such as bimanual-pot lifting and cloth folding.

### 4.3 Real Robot Experiments

In this subsection, we are interested in whether we can generalize to real world environments with RoboTube video dataset. We collect successful video episodes for each task under structured mode using a Franka robot with a Robotiq85 gripper. We us a fixed Intel D435 camera to record the videos from the third-person viewpoint. The models trained by self-supervised reward learning methods are used to predict the learned reward. Note that during video collection with the real robot, we also use unseen objects (i.e. objects in testing set). Under this overall setting, we aim to verify whether the

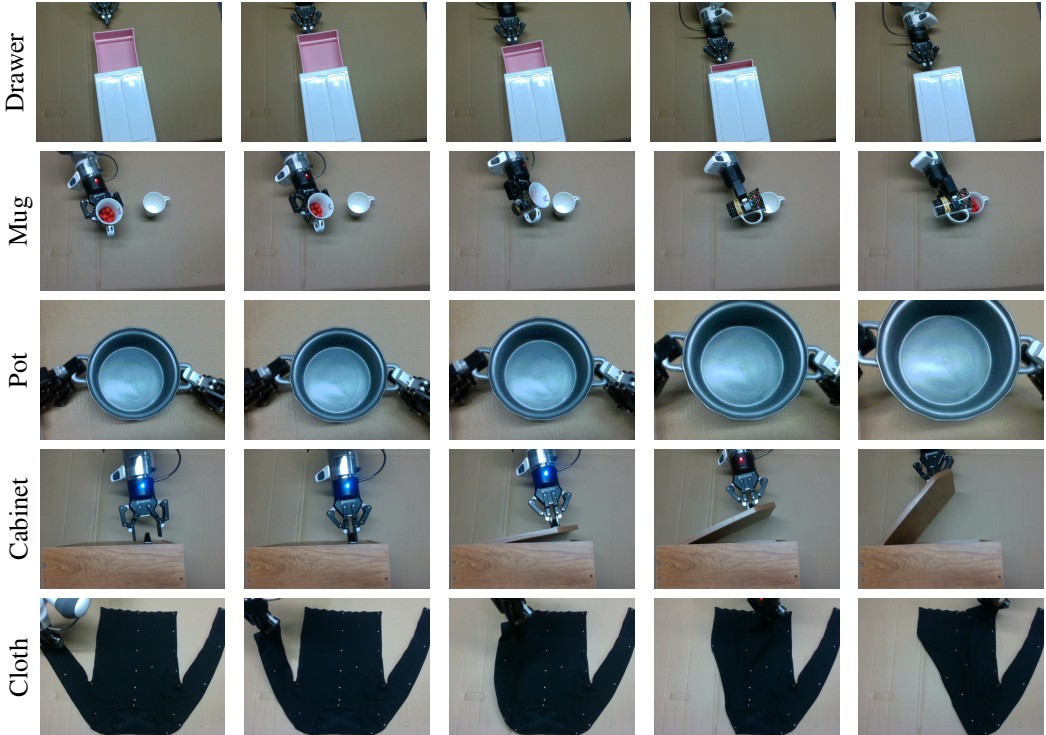

Figure 6: **Real robot experiments** We show the visual observations of the real robot for the five unseen objects from drawer closing task to cloth folding task.

reward models can generalize to 1) real-world setting where usually full of noise with unpredictable lighting conditions, 2) morphology gap that caused by the difference between the robotic end-effector and the human hand, and 3) unseen objects in the real world rather than only in the simulation.

As shown in Fig: 6. we can infer from the results that RoboTube has the potential to generalize to real robotic manipulation tasks. Hence it indicates that RoboTube can serve as a testing platform for helping researchers quickly validate the algorithms before testing on a real robot. Please find more details about the real world experiments in the appendix.

## 5   Limitations

Though we have already extended the robot learning from video tasks to a larger scope with complex task settings, diverse backgrounds, and object instances. And we also pay particular attention to asking the demonstrators to operate in a natural way. Our dataset still has a gap towards the ultimate "in-the-wild" setting where the videos from the internet can be much less structured or relevant.

## 6   Conclusion

We introduce RoboTube, a benchmark for robot learning from human videos. Our core contribution lies in the joint design of the RoboTube video dataset and RT-sim. The models learned from RoboTube videos can be tested, benchmarked, and reproduced in RT-sim. Extensive experimental results suggest that RoboTube can be served as a benchmark that guides the future development of robot learning from videos. Several potential future directions can be explored based on RoboTube: 1) The problem of in-the-wild human-to-robot imitation is an exciting direction. To step further in this direction, we plan to extend the data diversity of RoboTube to a larger scale. 2) RoboTube video dataset provides multiple features for self-supervised representation learning, which encourages more future algorithms in this direction. 3) Translating human videos into robotic demonstrations via explicit pose estimation combined with 3D vision is another interesting direction. 4) Learning to simulate by learning from observations. RoboTube has a collection of object models. It could be an exciting future direction of learning to simulate the objects from human videos.

**Acknowledgments**

Thank you Ziyi Wu, Samarth Sinha, Lixin Yang, Dr. Lin Shao, Dr. Liu Liu for the helpful discussions. We especially thank Dr. Huazhe Xu for the help of paper writing and discussions.

The research project is supported by Shanghai Qizhi Institute and MVIG lab at Shanghai JiaoTong University. We also thank CMU and Vector Institute for the computing resources.

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
