# OpenReview forum: "RoboTube: Learning Household Manipulation from Human Videos with Simulated Twin Environments"
_robot-learning.org/CoRL/2022/Conference — CoRL 2022 Oral_

### Official Review · Reviewer_wYij · 2022-07-16

**Originality:** Very Good
**Technical Quality:** Very Good
**Clarity Of Presentation:** Very Good
**Impact:** 4

**Recommendation:**

Weak Accept: I recommend accepting the paper, but will not argue for my recommendation if the majority of other reviewers have a different opinion.

**Summary:**

This paper presents a human household manipulation benchmark with associated digital twins and a focus on robot learning. The 5000 video clips cover tasks from articulated objects to deformable objects with multi-views and depth inputs, and the visual fidelity of the simulation is high and potentially useful for sim-to-real.

**Issues:**

The usage of unity simulator certainly provides better rendering quality, but the physics fidelity, especially contacts, are important for robotics and might not be as accurate. Moreover, I wonder about the speed of the simulation and the ease of usage for general robotic researchers.

I am not sure if I understand figure 5 correctly. Is the optimal reward function continuous and smooth along the time axis, and there should be a consistency for both real and sim? If so, it seems that none of the approaches are doing very well and no consistent win for either method? For instance, TCN seems to completely fail for simple tasks as cabinet, but works for complex task such as cloth folding. It’s interesting to think that in this case, simulation is used for test generalization and real-world data is used as training.

I also have confusions for table 2. It looks like that the designed environment reward is way much better than the learned rewards? I think this is reasonable and suggest more experiments on the representation or policy learning rather than inverse RL.

The details for the real-world experiments are not clear to me. What is the policy trained on, is it simulator with a learned reward? If the learned reward is good and the task is not that complicated, I am curious how well a simple MPC shooting method would work as a baseline.


**Quality Of The Limitations Section:**

Additional details required

**Reviewer Expertise:**

4: The reviewer is confident but not absolutely certain that the evaluation is correct

**Robotics Focus:**

Sufficient demonstration on hardware

**Strengths And Weaknesses:**

Strength:

The contribution of a large / diverse egocentric human manipulation video dataset has critical importance and good timing for the research community. Humans can provide many demonstrations for manipulation tasks and is a good way to scale up offline data.

Compared to existing benchmarks, depth and multi-view information are provided, which is important for 3D vision and spatial reasoning. Negative samples and simulated twins also open more research topics such as reinforcement learning in simulations, and avoid requiring excessive real-world experiments or only vision tasks for using this video benchmark.

The benchmark has a focus on robotics by design such as cluttered / clean scenes as well as simulating different robot arms and diverse poses, and its tasks are much more interesting than pick-and-place while not being too complicated as the full process of cooking a dish.

Weakness:

There are several arguments made in the paper but I think it would be great to have more convincing evidence.

The dynamics of simulations are claimed to resemble the real world but I can’t seem to find details for some system identification procedures during the benchmark curation. For instance, do all the shapes share similar physics parameters such as mass and frictions?
Similarly, to map the human arms to robot arms as well as the full scene, it seems some retargeting functions are needed and the human arms and some object pose / shape labels are needed, are these information also contained in the benchmarks and how accurate they are for cluttered scenes? It seems to me that Unity might have good content creation ability and rendering performance, but its physics engine might not have the enough fidelity to simulation contact-rich tasks in robotics.

A philosophical question. I find it interesting that the paper is taking a “real to sim to real” approach to tackle robotic tasks. I am curious about authors’ thoughts on the difficulty of building a world model compared to solving tasks for robotics. In other words, I agree that a simulator allows for benchmarking and democratizing researching progress, but are sim-to-real or real-to-sim problems necessary in the long run?
Another key usage of human demonstration datasets such as Ego4D is for representation pretraining, are there supporting results for better representation learned from a more focused robotic benchmark? Are there analysis of the multi-modality of the demonstrations?


**Summary Of Recommendation:**

This paper presents a novel and very relevant simulation and video benchmark for robotics. The designs such as multi-view, depth, task complexity, and cluttered scenes are important yet missing from benchmarks in other communities. There are still several weaknesses regarding experiments, missing details, and limitation discussions, and would love to hear about the response from the authors. Overall I recommend acceptance of this paper.

---

### Official Review · Reviewer_ZLYq · 2022-07-29

**Originality:** Very Good
**Technical Quality:** Very Good
**Clarity Of Presentation:** Very Good
**Impact:** 4

**Recommendation:**

Strong Accept: I recommend accepting the paper and will argue for my recommendation even if other reviewers hold a different opinion.

**Summary:**

The work presents a manipulation dataset consisting of two major parts. 1. Human demonstrated tasks, recorded via egocentric and stand mounted RGB cameras 2. A digital simulation of the tasks, called RT-sim. The dataset is intended for RL algorithms to achieve human-like manipulation with robotic systems. The paper demonstrates the application of some initial RL algorithms to the dataset.

**Issues:**

Minor weaknesses / typos
- Details of the robot used in section 5.3 would be welcome.
- Please expand the various acryonyms when first used. This was done for GC in section 5.1, but not AVID, LbW, TCN etc. This applies to other sections also. It is likely that readers will not be familiar with all acroynms and a general idea can be gained from such expansions without breaking the flow of reading.
- 'concert live' should be 'live concert'
- In the Mug pouring task description the authors write that there are '20 tiny balls'. Tiny is not very technical, and I would personally imagine something tiny to be around 1mm or less in diameter. Please remove the word tiny and include the diameter of the balls.

**Quality Of The Limitations Section:**

Limitations are addressed clearly

**Reviewer Expertise:**

4: The reviewer is confident but not absolutely certain that the evaluation is correct

**Robotics Focus:**

Sufficient demonstration on hardware

**Strengths And Weaknesses:**

The paper is on the whole well written and gives compelling arguments for the benefits of this dataset over existing datasets. There is some slight repetition here.

The tasks included in the dataset and interesting, though after watching the video supplement I was expecting there to be more tasks or object variations.

A strength is that the paper includes application of RL algorithms to the dataset, rather than simply saying that this is the intention of the work.

A major weakness is my opinion is the use of arbitrary objects. If the authors had used the YCB object set then users of the dataset would have been able to aquire physical objects in addition to digital models. Clearly the YCB object set would not address all of the tasks mentioned here (e.g. there is no drawer or cupboard), but it would have been nice to be able to use the same mug for testing on physical robots.



**Summary Of Recommendation:**

I recommend that this work be accepted for publication. I think the dataset will be a useful resource and the authors provide nice examples on its use. It is a pity that reproducable physical objects were not used.

---

### Official Review · Reviewer_fnSc · 2022-07-31

**Originality:** Good
**Technical Quality:** Fair
**Clarity Of Presentation:** Good
**Impact:** 3

**Recommendation:**

Weak Accept: I recommend accepting the paper, but will not argue for my recommendation if the majority of other reviewers have a different opinion.

**Summary:**

This paper presents a suitable dataset (RoboTube) for robot learning from videos. The dataset contains 5,000 video demonstrations recorded  with multi-view RGB-D cameras of human performing everyday household tasks. In contrast to existing video dataset for robot learning, RoboTube includes complex manipulation of different categories of objects such as rigid objects, articulated objects and deformable objects. The larger database of objects in the dataset promises to support a large range of diverse manipulation behaviours.
Robotube dataset is used to  evaluate a robot learning pipeline from human videos. To evaluate the learning process and benchmark different learning algorithm, a simulated twin environment  (RT-Sim) is developed. The simulator uses 3D scanned, photo realistic objects for minimizing the visual domain gap  between real and simulated environments.
The efficacy of the pipeline through sim to real transfer is validated on real experiments.

**Issues:**

Show the significance of  the proposed dataset with respect to others in Table 1 as how it addresses the deficiences mentioned in line 32-50.
Update: Satisfactory after revision

Also the significance of RT-Sim with respect to other simulators e.g., Robosuite, RLBench ...


**Quality Of The Limitations Section:**

Limitations are addressed clearly

**Reviewer Expertise:**

4: The reviewer is confident but not absolutely certain that the evaluation is correct

**Robotics Focus:**

Sufficient demonstration on hardware

**Strengths And Weaknesses:**

Strengths:
The paper addresses an important problem in robotics. The area and significance of research is very relevant to the robotics community. I agree that a suitable dataset for robot learning from human videos is missing and is needed.

The dataset has larger variety of object models and bigger database of videos. The simulated twin environment RT-Sim is able to compare different methods and also transfer the learning to real world.

Weakness:

The significance of the dataset with respect to the others in the state-of-the-art is not very clear. The comparison of features of the dataset with other existing ones on Table 1 does show that the dataset has larger database with diverse object models. However, the deficiencies mentioned is Line 32-50 is not well supported.
I do not agree that the manipulation tasks in RoboTube are long spectrum as compared to other dataset which is mentioned as short spectrum.
Also the data diversity and relevance of the RoboTube dataset is not measured using suitable metrics.
There already exists simulation environments like Robosuite and RLBench. I could not appreciate the development of  a new environment RT-Sim for benchmarking. I think either its significance is not well measured.
The link for the dataset pages is not updated. It would have been better to inspect the dataset.








**Summary Of Recommendation:**

I think the topic is interesting and a robotics focus dataset is developed for robot learning from videos.
However, I missed some comparative assessment in the paper and therfore its significance with respect to the state-of-the-art is not very clear.
I would want to see my concerns address for better recommendation.

---

### Meta-Review · Area_Chair_cMQo · 2022-08-13

**Recommendation:** Accept (Oral)
**Confidence:** 4

**Metareview:**

Summary:

In this paper, we propose a digital twin environment that builds a dataset of videos of humans performing household tasks as a resource for robots to acquire manipulation skills through learning and then uses the videos to simulate the actions in the videos. 5000 videos of human activities are used to show that effective learning can be achieved with real robots.

Strength:

The proposed dataset has a larger variety of object models and diverse egocentric human manipulation among 5000 videos.
The simulated twin environment RT-Sim can compare different methods such as reinforcement learning and transfer the learning to the real world.
Compared to existing benchmarks, the dataset has depth and multi-view information that is important for 3D vision and spatial reasoning. The benchmark focuses on robotics by design such as cluttered / clean scenes and simulating different robot arms and diverse poses.

Weakness:

The significance of the dataset with respect to the others in the state-of-the-art is not very clear.
The data diversity and relevance of the RoboTube dataset are not measured using suitable metrics. As there already exists simulation environments like Robosuite and RLBench, it is required to explain more carefully why the development of the RT-Sim is needed.
If the authors had used the YCB object set, then users of the dataset would have been able to acquire physical objects in addition to digital models. Some retargeting functions and the human arms and some object pose/shape labels are needed.

Feedback:

An important and relevant dataset MIME is missing. The advantage of RT_sim is the high-performance renderer which does not seem very significant. The high-fidelity object mesh models can be imported in similar Unity-based simulators. This can be attributed to an aspect of the dataset. RT-Sim advantages could have been tested with other simulators for the same task to prove its quantitative performance.


**Best Paper Nomination:**

No